# Clustering Mouse Movement Behavior in Surveys Using ResNet Embeddings

Lisa Bondo Andersen[1, 2], Tobias Wistuba[3], Felix Henninger[1, 2], Ailin Liu[1, 2], Sonja Greven[3], and Frauke Kreuter[1, 2]

[1]Ludwig Maximilian University of Munich
[2]Munich Center for Machine Learning
[3]Humboldt University, Berlin

## Abstract

Mouse movement trajectories in online surveys has been shown to reflect question difficulty during online surveys. We explore the use of deep neural network embeddings to summarize these trajectories, using a ResNet-based architecture applied to time-normalized cursor paths. Clustering and UMAP visualization of these embeddings on a subset of the data reveal a combination of large, dense clusters and smaller, distinct subgroups, suggesting diverse movement patterns among respondents. These preliminary findings indicate that neural embeddings can capture meaningful structure in survey interaction behavior, providing a foundation for further investigation into individual differences and adaptive survey design.

## 1  Introduction

Online surveys provide a scalable means to measure opinions, attitudes, and knowledge. However, unlike laboratory experiments, online surveys lack control of the environment of the respondent and, with it, the possibility to capture and respond to engagement, distractions, frustrations etc. respondents encounter [1]. Behavioral traces such as tracking the mouse movements and clicks of survey respondents can offer a richer view of the cognitive and motor processes that underlie responses [2, 3]. While mouse tracking has been used to infer hesitation, response uncertainty, and estimate workload [4, 5], less is known about how *personalized* these patterns are: do individuals exhibit consistent movement styles across different survey contexts? If so, is this effect consistent over time and perhaps across certain demographics?

We address these questions by applying deep learning methods to extract embeddings summarizing each respondent's cursor dynamics. By comparing feature spaces across survey tasks, we examine the degree of behavioral personalization and its relation to demographic and cognitive variables.

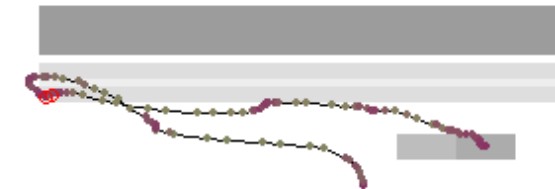

**Figure 1.** Input trajectory images for ResNet with color coding. Black (0, 0, 0) is used for the lines connecting the data points. Values between 50-149 indicate time (red channel), velocity (green channel), and acceleration (blue channel) of the data point. Values from 150-255 were reserved for visualizing the layout, and the red circles are clicks.

## 2  Methods

### 2.1  Data Collection

We collected mouse movement trajectories from Understanding America Study (UAS) respondents across three survey waves (January 2024, January 2025, and October 2025), encompassing questions on demographics, mental health, political opinions, and changes in lifestyle, employment, housing, and family composition. Emotional and political opinion questions further include self-reported difficulty levels on a 7 point scale. For each item, we recorded timestamped coordinates $(t, x, y)$, interaction events, and the underlying page layout.

### 2.2  Preprocessing

Using the `mousetrap` package [6], we time-normalized trajectories 101 samples per page, and derived cursor velocity, acceleration and distance traveled (among other features) at each timestep, producing a matrix $\mathbb{R}^{101 \times 3}$ representing position and dynamics. We then represented trajectories as images and the derived features via the RGB color channels, as visualized in figure 1. Each image represents the trajectory for one question for one respondent and served as inputs for the models.

## 2.3 Neural Network Architecture

We created two ResNet-based classifiers:

1. **Base ResNet-50:** We used a pretrained ResNet-50 architecture (trained on ImageNet) as a feature extractor by removing the final classification layer. The resulting feature embeddings from the penultimate layer were then used as input for the clustering analysis.

2. **Age-trained ResNet-50:** Starting from the same pretrained ResNet-50 backbone, we replaced the original classification head with a new one trained on our dataset to predict age classes. During fine-tuning, all ResNet layers were updated to adapt the network to the specific characteristics of our image data, which differ from those in the original ImageNet training set. After training, the classification layer was removed, and the fine-tuned feature embeddings were used for clustering.

The resulting embeddings $\mathbf{z}_i \in \mathbb{R}^d$ encode individual movement styles and serve as input for clustering and similarity analyses.

## 2.4 Clustering & Similarity Analysis

We applied unsupervised methods (k-means, UMAP visualization) to explore structure in the embedding space. Clustering quality was evaluated with respect to:

**Survey content:** Consistency across different question types (e.g. demographics, factual, behavioral, opinion questions).
**MouseTrap features:** Examine number and duration of hovers, horizontal and vertical flips, specific curves/movements across clusters.
**Individual differences:** Consistency of individual respondents, and, given this, how stable these patterns are over time (between survey waves).
**Demographics:** Distribution of demographic characteristics such as age, education level, and gender between clusters.
**Difficulty level:** Examining whether embeddings corresponding to similar self-reported task difficulty ratings cluster together.

## 3 Results

Preliminary analyses on a subset of the data are shown in Figure 2, which presents a two-dimensional UMAP embedding with 10 clusters.

The embedding exhibits a curved shape with two tails, where Clusters 4, 7, 9, and 0 are located, suggesting they represent less common patterns. Most points form dense clusters, with Clusters 1, 2, and 6 being particularly large, while Clusters 3, 5, and 4 are small or singleton clusters, highlighting rare

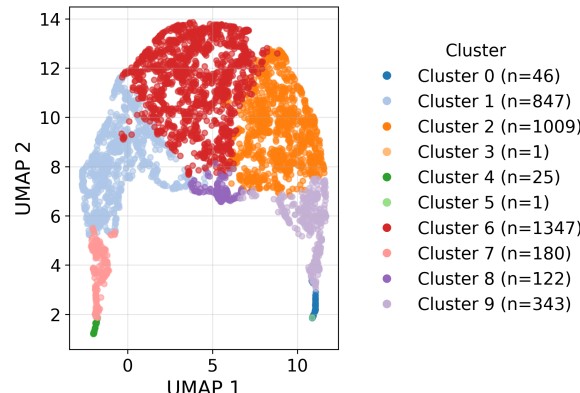

**Figure 2.** UMAP embedding of the subset of the data colored by 10 clusters.

or unique instances. Overall, the visualization indicates a combination of dominant patterns and diverse, smaller subgroups within the dataset.

## 4 Discussion and Conclusion

The UMAP embedding of ResNet-derived embeddings reveals both large, dense clusters and smaller, distinct subgroups, reflecting dominant behavioral patterns alongside rare or unique movement styles. Tails in the embedding, where Clusters 4, 7, 9, and 0 reside, indicate less common behaviors that warrant further analysis. Prior to the conference, we will extend the analysis to the full dataset, refine clustering parameters, and examine the characteristics of smaller clusters and outliers. Additionally, we will evaluate cluster quality against survey content, mouse-tracking features, individual consistency, and demographic variables, providing a more complete understanding of personalized behavioral signatures.

## Acknowledgments

We would like to thank our collaboration partners as well as the participants of the Understanding America Study for their valuable contributions. Funded by the Deutsche Forschungsgemeinschaft (DFG, German Research Foundation) – project number 396057129 ("Statistical modeling using mouse movements to model measurement error and improve data quality in web surveys").

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
