# OpenReview forum: "Clustering Mouse Movement Behavior in Surveys Using ResNet Embeddings"
_NLDL.org/2026/Abstracts_Track — NLDL 2026 Abstracts_

### Official Review · Reviewer_TGHQ · 2025-10-24

**Soundness:** 4
**Correctness:** 4
**Rating:** 5
**Confidence:** 3

**Summary:**

The authors investigate data which tracks mouse movements of survey participants, and aim to answer if there exist consistent movement patterns. They do so employing a resnet architecture, and achieved to find dominant patterns and subgroups.

**Strengths:**

- The problem at hand is clearly stated and motivated.
- The procedure is summarised clearly and in a coherent way.
- The results look promising.
- The next steps are outlined, promising to prompt an interesting discussion.
- The project applies machine learning techniques and constitutes an interesting application, hence this interdisciplinary work is relevant to the conference.
- Nice visualisations.

**Weaknesses:**

- The authors could motivate their choice of ResNets – is there a practical use? What has led to this decision? How doe sthis compare to other architectures?
- The baseline is missing: how does this method compare to others? This makes it hard to tell how much value the application of machine learning adds.
- The abstract does not include a summary of related work – this could be added.
- While the figures are nicely done, I do not fully understand Figure 1. Could the authors add a quick explanation how this mouse movement is supposed to be interpreted?
- The study touches upon many ideas and concepts, however it does not formulate a hypothesis. Of course an exploratory setting is natural for a work in progress, but in the future it would be valuable to know what you expect to uncover.

---

### Official Review · Reviewer_gsKU · 2025-11-04

**Soundness:** 2
**Correctness:** 2
**Rating:** 2
**Confidence:** 4

**Summary:**

This paper introduces a dataset capturing mouse movement behavior during a variety of online survey tasks. The dataset includes diverse samples and features (n = 101). The authors employ a ResNet-50 model for feature extraction and subsequently apply UMAP for dimensionality reduction and clustering (k = 10) to visualize the feature space.

**Strengths:**

1. The dataset is novel, providing diverse samples and feature representations relevant to behavioral modeling.
2. Additional feature engineering expands the potential analytical value of the data.
3. The proposed setup enables multiple promising research directions and downstream experiments.

**Weaknesses:**

1. Only a subset of the full dataset was analyzed.
2. Reliance on self-reported metadata introduces potential bias.
3. The paper does not clearly specify which features were used for training and testing the ResNet-50 model.
4. The rationale for using age as a prediction label is unclear.
5. The manuscript outlines five proposed experiments but reports no corresponding results.
6. The choice of 10 clusters in UMAP lacks justification or sensitivity analysis.

Minor issues:
1. Line 112: Present numbers in numerical order rather than clustering order for clarity.

---

### Decision · Program_Chairs · 2025-11-05

**Decision:**

Accept

**Comment:**

The reviewers found the abstract borderline, yet the PCs believe it will be of interest to the community and should have the opportunity be presented.